# *Aspergillus oryzae* Fermented Rice Bran: A Byproduct with Enhanced Bioactive Compounds and Antioxidant Potential

**DOI:** 10.3390/foods10010070

**Published:** 2020-12-31

**Authors:** Sneh Punia, Kawaljit Singh Sandhu, Simona Grasso, Sukhvinder Singh Purewal, Maninder Kaur, Anil Kumar Siroha, Krishan Kumar, Vikas Kumar, Manoj Kumar

**Affiliations:** 1Department of Food Science & Technology, Chaudhary Devi Lal University, Sirsa 125055, India; siroha01@gmail.com (A.K.S.); k.kumar4032@gmail.com (K.K.); vk.pandit415@gmail.com (V.K.); 2Department of Food Science & Technology, Maharaja Ranjit Singh Punjab Technical University, Bathinda 151001, India; purewal.0029@gmail.com; 3Institute of Food, Nutrition and Health, University of Reading, Reading RG6 6UR, UK; simona.grasso@ucdconnect.ie; 4Department of Food Science & Technology, Guru Nanak Dev University, Amritsar 143005, India; mandyvirk@rediffmail.com; 5Chemical and Biochemical Processing Division, ICAR—Central Institute for Research on Cotton Technology, Mumbai 400019, India

**Keywords:** rice bran, solid state fermentation, antioxidant activity, bioactive compounds, *Aspergillus oryzae*, HPLC, total phenolic content, reducing power assay

## Abstract

Rice bran (RB) is a byproduct of the rice industry (milling). For the fermentation process and to add value to it, RB was sprayed with fungal spores (*Aspergillus oryzae* MTCC 3107). The impact of fermentation duration on antioxidant properties was studied. Total phenolic content (TPC) determined using the Folin–Ciocalteu method, increased during fermentation until the 4th day. The antioxidant activity analyzed using the 2,2 Diphenyl–1′ picrylhydrazyl (DPPH) assay, total antioxidant activity (TAC), 2,2′-azinobis 3-ethylbenzothiazoline-6-sulfonic acid (ABTS^+^) assay, reducing power assay (RPA) and hydroxyl free radical scavenging activity (HFRSA) for fermented rice bran (FRB) were determined and compared to unfermented rice bran (URB). TAC, DPPH, ABTS^+^ and RPA of FRB increased till 4th day of fermentation, and then decreased. The specific bioactive constituents in extracts (Ethanol 50%) from FRB and URB were identified using high performance liquid chromatography (HPLC). HPLC confirmed a significant (*p* < 0.05) increase in gallic acid and ascorbic acid. On the 4th day of fermentation, the concentrations of gallic acid and ascorbic acid were 23.3 and 12.7 µg/g, respectively. The outcome of present investigation confirms that antioxidant potential and TPC of rice bran may be augmented using SSF.

## 1. Introduction

Rice (*Oryza sativa*) belongs to the grass family and is the most widely consumed grass by a significant proportion of human population, especially in Asian regions. It is an agricultural commodity with the third highest worldwide production [1]. The total worldwide production of rice was about 769,657,791 tonnes in an area of 167,249,103 ha, of which India produced 168,500,000 tonnes [1]. Rice bran (RB) is the major byproduct of milling industry, especially processing rice, and ultimately represents 5–10% of the total grain. RB constitutes crude protein (11–13%), oil (20%) and dietary fibers (22.9%), including hemicelluloses, arabinogalactan, arabinoxylan, xyloglycan, and raffinose with good sources of bioactive ϒ-oryzanol, Vitamin-E and minerals [2,3,4].

In routine practice, RB is used as feed for animals or in the production of edible cooking oils [5]. In the context of making our economies more circular and our diets more sustainable, there is a growing need and interest to valorize byproducts into new sustainable food ingredients with high nutritional value. Fungal fermentation is a promising method to process agricultural byproducts and to produce value added products [6]. SSF usually starts with the growth of fungal strains on substrate with little or no free water, with several advantages, including low costs, low environmental impact and high reproducibility [7].

Scientific reports supporting effect of fermentation on the antioxidant levels of various substrates, including barley [8], pearl millet [9,10], wheat [11], and rice bran [12,13], and reported their enhancement after SSF. This is a commonly used approach by the scientific community for the improvement of bioactive content of agro-industrial residues and assisted in reducing the environmental pollution caused by these residues. It is also evident from the findings that SSF may be used to improve product functional properties and as a tool to develop cereals with beneficial nutritional properties. SSF using *Rhizopus oligosporus* and *Monascus purpureus* enhanced the quality of fermented RB in terms of antioxidant property and total phenolic compounds [12]. Authors achieved maximum antioxidant capacity (more than 5-fold compared to untreated RB), and total phenolic content (more than 8-fold compared to untreated RB), when RB was fermented with the mixed cultures of *R. oligosporus* and *M. purpureus.* The other strain, *Rhizopus oryzae*, was also investigated by another group of researchers who established that SSF using *R. oryzae* improves the overall nutritional profile of the RB with excellent antioxidant activities [13]. 

The RB fraction has not been as much of a focus of research compared to polished rice. RB has well known for health beneficial properties due to its bioactive compounds, and during the present experimental work, an attempt was made to further enhance the antioxidant content of RB using *Aspergillus oryzae* as a starter culture. *Aspergillus oryzae* in particular was used because fungal strains, especially those belonging to the *Aspergillus* group, are well known for their potential to produce hydrolytic enzymes which resulted in enhanced production of bioactive compounds, especially cinnamic acids in fermented substrates during SSF [14]. SSF could be an important process to prepare antioxidant rich products with industrial applications. This process is comparatively cheaper than any other method of modulating nutrients. Furthermore, the efficacy of fungal strains towards the improvement of nutrients may vary with the substrate nutritional profile. Using SSF, those substrates could also be processed in the form of antioxidant rich food/feed which initially considered as waste. *Aspergillus oryzae* is widely used for fermentation of different natural resources such as rice [15,16]; brown rice and rice bran [17]. *Aspergillus oryzae* is a famous fungus commonly used for the preparation of local foods and beverages in Japan for the preparation of sweet potato, sake, shōchū, soy sauce and miso. Hence, it is evident that use of *Aspergillus oryzae* is common practice aiming to produce foods with high nutraceutical values. This is the first study investigating the use of SSF on RB using *Aspergillus oryzae* to evaluate the effect on antioxidant properties and bioactive compounds. 

## 2. Experimental Details

### 2.1. Chemicals

Organic solvents and chemical reagents such as catechin, gallic acid, 2,2-diphenyl-1-picrylhydrazyl (DPPH), 2,2′-azinobis 3-ethylbenzothiazoline-6-sulfonic acid (ABTS^+^), ascorbic acid used were of analytical grade and procured from HiMedia and Sigma-Aldrich. HPLC grade standards were procured (HiMedia, India) and used during HPLC analysis for estimating specific compounds in RB extracts. The glassware used in the present experimental part was of Borosilicate. Before using, glassware was washed with Labolene detergent and rinsed with tap water and sterilized in an oven at 100 °C for 1.5 h.

### 2.2. Isolation of Rice Bran (RB)

The experimental sample (Paddy cultivar PB-1121) was obtained from a local market in Sirsa, India. Sample grains were washed, dried and stored in airtight containers. A paddy dehusker (Khera, Delhi, India) was used for de-husking the paddy and a rice polisher (Khera, Delhi, India) was used to separate the bran. RB was converted to a powdered form using mixer-grinder (Bajaj, India) and stored in deep freezer (−20 °C; Vestfrost, India).

### 2.3. Starter Culture for Solid State Fermentation (SSF)

Fungal culture (*Aspergillus oryzae* MTCC 3107) for fermentation of RB was procured. Starter culture was grown on CYEA media (czapek yeast extract agar) and CYEB (czapek yeast extract broth) at 25 ± 2 °C. Steam sterilized RB was inoculated by spraying spores as a suspension (2 mL, 1 × 10^5^). RB sample which was not sprayed with fungal spores assigned name as URB.

### 2.4. SSF of RB

The experimental sample (50 g powdered RB) was used as substrate in Erlenmeyer flasks (250 mL). The sample was soaked in CYB (czapek yeast broth, 1:1 *w*/*v*) at ambient conditions for 10–12 h. Substrate was sprayed with the spores suspension and incubated (7 days, 25 ± 2 °C). Choice and types of media used for starter culture growth during SSF merely vary with the fungal strain or starter culture type. Fermented rice bran (FRB) was removed from flasks after a predetermined interval of time (24 h) and dried in an oven (Narang Scientific Works, New Delhi, India) at 45 °C (24 °C 48 h). Fermented and unfermented substrate was converted to flour (Sujata, India). URB and FRB flour was defatted with hexane (1:5 *w*/*v*, 3 times, 5 min), dried in an oven (NSW, India) and extracted with organic phase ethanol (50%) at 45 °C for 30 min. Before performing the extraction process, flour samples were sieved to attain uniform sized particles for extracting bioactive compounds from them. 

### 2.5. Evaluation of Phytochemical Composition

The detection of specific phytochemicals in the URB and FRB extracts was carried out using various qualitative tests as per standard methods described by [18]. Different tests were performed to assess the chemical composition of URB and FRB, such as saponin, steroids, flavonoids, coumarins and alkaloids.

#### 2.5.1. Total Phenolic Content (TPC)

TPC in FRB and URB extracts was estimated by following FCR (Folin–Ciocalteu reagent) method [19]. An aliquot (100 μL) of extract was allowed to react with FC reagent (500 μL) and after an incubation period of 5 min, sodium carbonate (1500 μL) was added to the reaction mixture and total volume (10 mL) was prepared with distilled water. Absorbance was recorded at 765 nm against a blank. The standard used during the TPC assay was gallic acid. 

#### 2.5.2. Determination of Saponins

An amount of 5 mL of water was kept in storage vial following the addition of 1 mL of URB and FRB extract and the tube was shaken vigorously. The formation of lather confirms saponin presence.

#### 2.5.3. Determination of Steroids

To detect steroids, 2 mL of URB and FRB extracts were taken, and 2 mL of chloroform was added followed by 2 mL of conc. H_2_SO_4_, red color in the chloroform layer showed the presence of steroids.

#### 2.5.4. Determination of Flavonoid

To detect flavonoids in URB and FRB, 1 mL of 10% Lead Acetate was added to 1 mL of URB and FRB extracts; the formation of yellow-colored precipitates is an indication for the presence of flavonoids.

#### 2.5.5. Condensed Tannin Content (CTC)

Quantification of CTC in URB and FRB extracts was calculated by vanillin: HCl protocol [20]. Aliquot of URB and FRB extracts (100 μL) was taken separately in storage vial (5 mL) followed by Vanillin–HCl (1:0.5) addition. The mixture was incubated (ambient temp. for 10 min) and absorbance was recorded.

#### 2.5.6. Determination of Coumarins

To determine the presence of coumarins 1 mL of 10% NaOH was added to 1 mL of URB and FRB extracts. The formation of yellowish color showed the presence of coumarins.

#### 2.5.7. Alkaloids

Alkaloids in URB and FRB extracts were estimated using three different tests.

Wagner’s test: URB and FRB extracts (2 mL) were treated with Wagner’s reagent (2 mL), the formation of precipitate (reddish-brown) confirmed alkaloids in sample extract.Mayer’s test: To URB and FRB extract (1 mL), Mayer’s reagent (2 mL) was added, and precipitate (dull white) confirmed alkaloids presence in sample extract.Hager’s test: To URB and FRB extract (1 mL), Hager’s reagent (3 mL) was added, and the formation of precipitate (yellow) confirmed alkaloids in sample extract.

#### 2.5.8. Qualitative and Quantitative High Performance Liquid Chromatography (HPLC) Analysis

HPLC analysis of extracts (URB and FRB) was performed as per an already published report on antioxidants [21]. Extracts were prepared using the already published report [22]. HPLC analysis of URB and FRB extract for the estimation of bioactive compounds was performed (Shimadzu 10 AVP HPLC system). HPLC is an important step during analysis of extracts at different levels as it helps in validating the specific effects and finalizing the concept related to specific process. For the HPLC analysis of extracts and determination of specific bioactive compounds in them, a Shimadzu 10 AVP HPLC system was used which comprises SCL10 AVP system controller and two pumps (LC-10 AVP) CTO-10 AVP column oven with injection (Rheodyne 7120) value (20 µL sample loop) and photodiode-array detector (SPD-M10 AVP). Analytical HPLC column (Gemini-NX C18) (250 × 4.6 mm, 3 µm) with a guard column (40 × 3 mm, 3 µm) both from Phenomenex (Torrance, CA, USA) was used. Experimental performance was conducted at a rate (0.5 mL/min) using acetic acid (1.5% *v*/*v* solvent A) and aqueous ethanol:acetonitrile (40:50 *v*/*v*) mixture (solvent B) under the following gradient program: 0–8 min. 70% acetic acid, 8–19 min. 60% acetic acid, and 19–30 min. 30% acetic acid. Injection volume was 10 µL. The analysis was completed with different chromatograms formation at 280 nm.

### 2.6. Assessment of Antioxidant Properties in URB and FRB

#### 2.6.1. DPPH (2,2-Diphenyl–1′ picrylhydrazyl) Assay

Detection of DPPH scavenging activity in URB and FRB extracts was estimated [23]. First, 100 µL of URB and FRB extract was added (test tubes) followed by addition of 3 mL of DPPH (100 µM). Absorbance of URB and FRB extracts treated with DPPH solution was taken at 517 nm after 30 min of reaction process. Formula for calculation of percent (%) DPPH inhibition is mentioned below
Percent (%) DPPH inhibition = (A_C_ − A_E_/A_C_) × 100(1)
A_C_ (absorbance of control); A_E_ (absorbance of extracts).

#### 2.6.2. ABTS Assay

Scavenging of the ABTS solution by URB and FRB extracts was calculated [24,25]. During the ABTS assay, URB and FRB extracts were allowed to react with potassium persulfate treated ABTS solution (16 h incubation). After 10 min of reaction time, the absorbance was taken (732 nm). The assay helps to study how extracts behaves under the oxidative stress conditions formed during normal biological processes. Percent (%) ABTS inhibition was calculated as mentioned below:Percent (%) ABTS inhibition = (A_C_ − A_E_/A_C_) × 100(2)
A_C_ (absorbance of control); A_E_ (absorbance of extracts).

#### 2.6.3. HFRSA Assay

The radical scavenging potential of URB and FRB extracts was estimated [26].
Scavenged OH % = [(A_C_ − A_E_)/A_C_ × 100](3)
A_C_ (absorbance of control); A_E_ (absorbance of extracts).

#### 2.6.4. Total Antioxidant Capacity (TAC)

The antioxidant capacity of extracts during TAC assay was determined by following the method as described by Prieto et al. [27]. URB and FRB extracts were analyzed for antioxidant capacity using sodium hydrogen orthophosphate (28 mM); conc. H_2_SO_4_ (0.6 M) and ammonium molybdate (4 mM) at 95 °C for 90 min. 

#### 2.6.5. Reducing Power Assay (RPA)

RPA of extracts (URB and FRB) was measured as per standardized method [28]. URB and FRB extract (100 μL) was allowed to react with aqueous potassium ferricyanide solution (1%; 100 μL) in water bath (50 °C for 30 min) and after reaction in water bath trichloroacetic acid (1% TCA; 100 μL) was added following incubation under dark (15 min). Dilution with double distilled water was done to achieve final volume (10 mL). Absorbance of colored complex was recorded at 700 nm. Activity was measured against quercetin (standard).

### 2.7. Statistical Analysis

Triplicate observations were processed through ANOVA using Minitab software (Version 16, Minitab Inc., State College, PA, USA).

## 3. Results and Discussion

### 3.1. Effect of SSF on Phytochemicals and TPC

Preliminary screening was carried out for URB and FRB to detect the presence of phytochemicals and the results of phytochemical analysis are shown in Table 1. 

Result shows the presence of coumarins and sugars in the extracts of samples. Shahidi [29] stated that phenolic and poly-phenolic compounds include a main class of secondary metabolites that act as free radical scavengers, reducer of low-density lipoprotein (LDL) and oxidation of cholesterol. TPC of URB was found to be 1.08 mg gallic acid equivalents (GAE)/g; significant (*p* < 0.05) difference among rice bran fermented for different durations was observed (Table 2). 

Increase in TPC was observed till the 4th day of fermentation, and further increase in time of fermentation TPC was decreased. On 4th day of fermentation, TPC was 8.83 mg GAE/g and the percentage increase in TPC content in FRB as compared to URB was 717%. Significant enhancement in amount of TPC during fermentation is a strong indication of the positive effect of SSF on substrate as later on antioxidant properties are solely dependent on the amount and type of phenolics. Fungal fermentation is considered as important phenomenon as desirable changes in nutritional profile could be achieved within short span of time. Schmidt et al. [13] observed two-fold increments of TPC in RB after SSF using *R. oryzae*. The increase in TPC till the 4th day may be due to hydrolytic enzymes produced during SSF [10]; however, the degradation of gallic acid to aliphatic compounds might be responsible for the decrease afterwards [30].

### 3.2. Effect of SSF on CTC

CTC is important as other secondary metabolites in the food. They have high antioxidant activity in vitro compared to monomeric phenolic compounds [31]. Significant (*p* < 0.05) differences were observed in CTC between URB and FRB and the values ranged between 34.6 to 365 mg CE/100 g extract (Table 2). An increase in CTC of FRB was observed till the 3rd day of fermentation, and thereafter the reverse was observed. Up to 10-fold increases with a percentage increase of 952% were observed on 3rd day of fermentation. Releases of enzymes take place after fermentation, which results in the production of plant chemicals such as tannin, alkaloids and phenylpropanoids.

### 3.3. Effect of SSF on Specific Bioactive Constituents

Phenolic compounds increase their antioxidant activity by various methods [32] and the effectiveness of these phenolic compounds as antioxidants mainly depends on their chemical structures, relative orientation, and number of OH groups attached to the aromatic ring [33]. Qualitative and quantitative measurements for the detection of specific phenolic compounds in URB and FRB were determined using HPLC. Standards *viz.* gallic acid, ascorbic acid, catechin and vanillin were used to evaluate the bioactive compounds (phenolic acids) in both URB and FRB. Bioactive compounds present in URB were significantly modulated during SSF which was also confirmed during HPLC analysis. The changes in the amount and type of bioactive compounds depend on enzymatic activity during the fermentation process. The presence of specific compounds in fermented products also makes them important substrate for pharmaceutical industries as it could also be used in preparation of various health benefiting formulations. The results of bioactive compounds in URB and FRB extracts are shown in Table 3 and Figure 1 and Figure 2.

Results from the quantitative analysis of sample showed that the quantity of identified bioactive compounds varied from 1.2 to 23.3 µg/g, respectively. The composition of phenolic acid was significantly (*p* < 0.05) influenced by the duration of SSF. The effect of SSF on bioactive profiles depends on the type of substrate, starter culture and the extraction conditions after SSF [9,34]. After SSF, an increase in gallic acid (23.3 µg/g) was observed, followed by ascorbic acid (12.7 µg/g). Catechin (5.8 µg/g) and vanillin (1.2 µg/g) decreased after 4 days of fermentation. The reduction in catechin and vanillin may be due to the degradation of these compounds by microflora. The results are in line with the other researchers, where a significant (*p* < 0.05) enhancement in ferulic, sinapic, vanillic, caffeic, syringic, and 4-hydroxybenzoic acids of RB fermented with *Rhizopus oligosporus* and *Monascus purpureus* [12]. The authors also reported that the amount of ferulic acid was 8-fold higher compared to the untreated RB samples, suggesting the novelty of SSF for enhancing the phytochemical content. The increased phenolic acid content in RB is mainly caused due to the cleavage of compounds in conjunction with lignin [35]. In another study, it was reported that SSF using *Rhizopus oryzae* resulted in a 110% improvement in the phenolic compounds content [13]. The content of vanillin, chlorogenic acid, and p-hydroxybenzoic acid were increased throughout the fermentation process. The highest increment was detected in the ferulic acid (764.7 mg/g on dry weight basis) after SSF of 120 h. Authors concluded that *R. oryzae* produces certain enzymes which degrade the rigid cell wall of the RB and resulted in the release of the ferulic acid [13]. The increment in total phenolic content, and antioxidant properties may also attribute to the hydrolytic enzyme present in the fungal strain. These enzymes act upon the substrate (RB) and increase the access of hydroxy functional groups on the phenolic compounds. This improves the number of phenolic groups and, as a result, improves the antioxidant properties of the treated sample [12]. It is evident from the results that fermented Rb can be an important functional ingredient in the development of innovative food items with high phenolic content and antioxidant properties. 

### 3.4. Effect of SSF on Antioxidant Activity

Evaluating the scavenging activity is a critical process, and a number of analytical methods may be used to evaluate the antioxidant activity of sample prepared from natural resources [36,37]. The DPPH radical scavenging assay reflects the capacity of the extract to transfer electrons or hydrogen atoms whereas ABTS^+^ radical scavenging activity shows the hydrogen donating and the chain-breaking ability of the extract [38]. DPPH is popularly used as choice assay for analyzing the antioxidant properties of plant samples in a short period of time compared to other antioxidant assays. For URB, the free radical scavenging capacity of the sample using DPPH and ABTS^+^ was observed to be 75.4 and 35.3%, respectively (Table 4).

Scavenging activities were increased till the 4th day of fermentation and further increase in fermentation time reverse was observed. The values observed on 4th day were 85.4% and 82.7% with percentage increase of 12.6 and 136.5% for DPPH and ABTS+ tests, respectively. Fermentation showed a positive effect on DPPH and ABTS^+^ inhibitory effect on RB. Increase in antioxidant characteristics may be due to an increase in phenol and anthocyanin contents during fermentation [39]. Belefant-Miller et al. [40] suggested that the DPPH radical-scavenging activity of FRB was extremely correlated to the presence of antioxidant secondary metabolites. A significant increase in TPC was observed after the fermentation of mung bean cultivars [41]. The antioxidant activity of FRB was observed to be higher compared to URB when the HFRSA method was chosen for evaluating the antioxidant activity, and the values varied from 13.3¨ to 28.8%, respectively (Table 4). TAC is the total capacity of antioxidants for removing the free radicals in cells [42]. As shown in Table 4, the highest TAC was observed on the 4th day of fermentation of FRB sample as compared to the URB. The values for TAC ranged between 7.3 mg/g (URB) and 15.4 mg/g (FRB). TAC increased until the 4th of fermentation and subsequently declined on the 5th day. The release of flavonoids during the fermentation process enhances the anti-oxidative activity from plant-based foods, which may be a useful method of improving the supply of natural antioxidants [42]. The reducing power (RP) is a main indicator of the potential antioxidant activity of antioxidants. The antioxidant effect statistically increases as a function of the development of the RP, indicating that the antioxidant properties are related to the development of RP [28]. The highest RPA (16.5 mg/g) was observed on the 4th day of incubation of FRB extracts as compared to the URB (0.75 mg/g) (Table 4). A similar increase in RPA of rice fermented by *Phellinus linteus* was observed by Liang et al. [43]. During fermentation, phenolic content increased and these compounds can act as reducing agents and hydrogen donors [44]. Study of fermented products in different aspects proved to be helpful as it clearly demonstrates SSF effect whether positive or negative. Further, detailed analysis of fermented products helps to eradicate the negative aspects. As nutrients present in fermented products could be capable of combatting various medical problems and, hence, these products could be recommended to persons suffering from certain specific disorders.

## 4. Conclusions

The SSF of RB with *Aspergillus oryzae* significantly (*p* < 0.05) increased the TAC, TPC, CTC, RPA, DPPH, ABTS^+^ and HFRSA. Except for CTC, the bioactive properties studied showed the maximum increase until 4th day of fermentation. Increase in SSF duration after a specific period resulted in the loss of important bioactive constituents. Four standards *viz.* gallic acid, ascorbic acid, catechin and vanillin were chosen to screen bioactive compounds using HPLC. SSF using *Aspergillus oryzae* thus can be effective method for the increment of antioxidants in rice bran. More research is required to optimize the size of inoculum, starter culture age and extraction parameters. The results of this study demonstrate that fermented rice bran would be an antioxidant rich and healthy food supplement as compared to non-fermented rice bran. After evaluating of the other nutritional components (shelf life and sensory analysis) under standardized conditions, the fermented rice bran could be used for formulation of different health benefiting food products. The SSF process could be recommended to modulate the bioactive profile and nutritional composition of industrial waste as well as eatable food materials. More research is also needed to standardize the fermentation process using other microorganisms so that their effect on nutritional quality may be evaluated.

## Figures and Tables

**Figure 1 foods-10-00070-f001:**
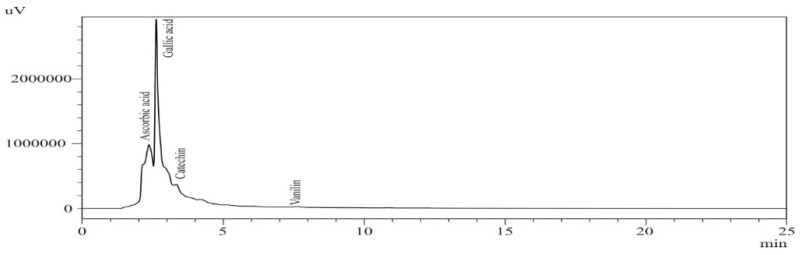
Phenolic acid composition of un-fermented rice bran at 280 nm.

**Figure 2 foods-10-00070-f002:**
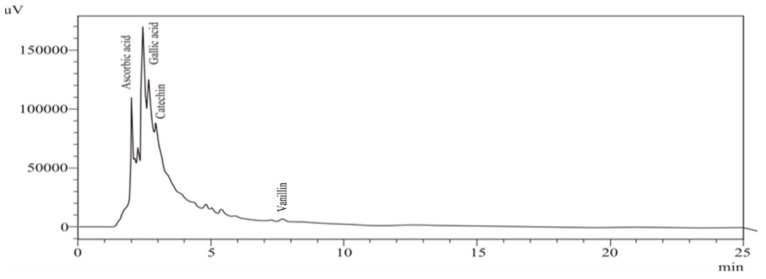
Phenolic acid composition of fermented rice bran (4th day) at 280 nm.

**Table 1 foods-10-00070-t001:** Chemical composition of unfermented rice bran (URB) and fermented rice bran (FRB) (4th day) extracts.

Phytochemical	URB	FRB (4th Day)
Coumarins	+	+
Flavonoids	-	-
Saponin	-	-
Steroid	-	-
Alkaloids	-	-

(+) present whereas (-) sign showed absence.

**Table 2 foods-10-00070-t002:** Impact of time of fermentation on total phenolic content (TPC) and condensed tannin content (CTC) of rice bran (RB) extracts.

Fermentation Time (Days)	TPC (g GAE/g dwb)	Percent (%) Change in TPC after SSF	CTC (mg CE/g dwb)	Percent (%) Change in CTC after SSF
URB	1.08 ± 0.13 ^a^	--	34.6 ± 0.06 ^a^	--
1	4.15 ± 0.09 ^b^	_↑_284%	261 ± 0.05 ^d^	_↑_652%
2	5.11 ± 0.10 ^c^	_↑_373%	227 ± 0.07 ^c^	_↑_555%
3	6.52 ± 0.11 ^e^	_↑_503%	365 ± 0.13 ^g^	_↑_952%
4	8.83 ± 0.21 ^g^	_↑_717%	295 ± 0.11 ^f^	_↑_750%
5	6.96 ± 0.34 ^f^	_↑_544%	269 ± 0.02 ^e^	_↑_675%
6	6.37 ± 0.19 ^d,e^	_↑_489%	268 ± 0.07 ^e^	_↑_674%
7	5.86 ± 0.08 ^d^	_↑_442%	159 ± 0.09 ^b^	_↑_358%

GAE: Gallic acid equivalent; SSF: Solid substrate fermentation; TPC: Total phenolic content; CTC: Condensed tannin content. Means followed by same superscript within a column do not differ significantly (*p* < 0.05). Subscripts show the % increase (↑) from unfermented sample for corresponding properties.

**Table 3 foods-10-00070-t003:** Phenolic acid composition of un-fermented and fermented rice bran (4th day).

Compounds	URB	FRB (4th Day)
Ascorbic acid (μg/g)	11.1 ^a^	12.7 ^b^
Gallic acid (μg/g)	14.8 ^a^	23.3 ^b^
Catechin (μg/g)	9.6 ^b^	2.8 ^b^
Vanillin (μg/g)	5.8 ^a^	1.2 ^a^

Means followed by similar superscript within a column do not differ significantly (*p* < 0.05).

**Table 4 foods-10-00070-t004:** Effect of duration of fermentation on DPPH, ABTS inhibition, total antioxidant capacity (TAC), hydroxyl free radical scavenging activity (HFRSA) and reducing power assay (RPA).

Fermentation Time (Days)	DPPH (% Inhibition)	ABTS (% Inhibition)	TAC (mg AAE/g dwb)	HFRSA (% Inhibition)	RPA (mg QE/g dwb)
URB	75.4 ± 0.11 ^a^	35.3 ± 0.48 ^a^	7.3 ± 0.46 ^a^	13.3 ± 0.90 ^a^	0.7 ± 0.18 ^a^
1	77.8 ± 0.33_↑__3.08_ ^e^	75.8 ± 0.89_↑__116.5_ ^d^	9.7 ± 0.32_↑__31_ ^b^	28.8 ± 0.42_↑__116_ ^e^	2.7 ± 0.39_↑__260_ ^b^
2	78.5 ± 0.24_↑__4.12_ ^d,e^	78.5 ± 0.77_↑__123.5_ ^e^	13.5 ± 0.38_↑__87_ ^c^	25.3 ± 0.66_↑__90_ ^c^	3.5 ± 0.16_↑__368_ ^c^
3	83.1 ± 0.20_↑__10.12_ ^f^	79.8 ± 0.54_↑__127.9_ ^f^	14.7 ± 0.19_↑__99_ ^d,e^	25.9 ± 0.48_↑__94_ ^c^	8.5 ± 0.22_↑__1040_ ^e^
4	85.4 ± 0.23_↑__12.66_ ^g^	82.7 ± 0.71_↑__136.5_ ^g^	15.4 ± 0.24_↑__103_ ^f^	28.5 ± 0.24_↑__114_ ^e^	16.5 ± 0.24_↑__2102_ ^g^
5	77 ± 0.19_↑__2.09_ ^d^	75.2 ± 0.85_↑__114.6_ ^d^	14.8 ± 0.61_↑__101_ ^e^	26.1 ± 0.56_↑__96_ ^d^	12.6 ± 0.21_↑__1589_ ^f^
6	76.2 ± 0.56_↑__1.046_ ^c^	70.7 ± 0.42_↑__101.9_ ^c^	14.5 ± 0.53_↑__97_ ^d^	24.7 ± 0.53_↑__85_ ^c^	8.8 ± 0.27_↑__1082_ ^e^
7	75.5 ± 0.17_↑__0.066_ ^b^	69.5 ± 0.23_↑__98.6_ ^b^	13.9 ± 0.39_↑__88_ ^c^	20.2 ± 0.85_↑__51_ ^b^	6.9 ± 0.34_↑__826_ ^d^

DPPH—2,2-diphenyl-1-picrylhydrazyl; ABTS-2,2′—Azinobis(3-ethylbenzothiazoline-6-sulphonic acid) diammonium salt; TAC-Total Antioxidant Capacity; HFRSA—Hydroxyl Free Radical Scavenging Activity; RPA—Reducing Power Activity. Means ± standard deviation, values followed by similar superscript within a column do not differ significantly (*p* < 0.05). Subscripts denote the percentage increase (↑) from unfermented sample for corresponding properties.

## Data Availability

Not Applicable.

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
