# Peer review of "Aspergillus oryzae Fermented Rice Bran: A Byproduct with Enhanced Bioactive Compounds and Antioxidant Potential"

_foods, 2020, doi:10.3390/foods10010070_

Round 1
Reviewer 1 Report
The manuscript entitled “Aspergillus oryzae Fermented Rice Bran: A By-Product with Enhanced Bioactive Compounds and Antioxidant Potential” authored by Sneh Punia and colleagues deals with the investigation of fermentation processes operated by Aspergillus oryzae on Oryza sativa bran. In particular, the authors evaluated if the fermentation process could affect the phytochemical profile and the antioxidant activities of the raw material during a limited time period (4 days).
The paper may contain partially interesting data, however it is really badly organized. Furthermore, I have strong doubts about the methodologies used by the authors.
INTRODUCTION:
The introduction is well written. The authors elaborate with particular regard the Rice Bran's economic impact, however:
- no information on the use of Aspergillus oryzae is provided. Aspergillus oryzae is a famous fungus commonly used for the preparation of local foods and beverages in Japan (for example for the preparation of sweet potato, sake, shōchū, soy sauce and miso).
- Moreover, information regarding other food products based on Rice but produced with other kind of inoculation (for example, fermented red rice) should at least be mentioned, underlining that the methodology operated by the authors is a very common practice aimed at the production of foods with high nutraceutical values.
MATERIALS AND METHODS:
Section 2 should be renamed: Materials and Methods. This section is really confusing. Materials and methods are poorly described, and some paragraphs are not at all.
- Subsection 2.5.1. is not clear. What authors state to evaluate in this subsection? No methodology is described in this section.
- Moreover, the authors confuse the phytochemical compounds (polyphenols, alkaloids, etc.) deriving from the secondary metabolism of plants, from those deriving from the primary metabolism (sugars, fats, fibres, proteins, minerals). The latter compounds may be included as proximal composition of foods, and not in the phytochemical profile. If all these analyses have been performed, authors should separate them into two different sections (one concerning phytochemical evaluations, and the second one on proximal composition).
- as far as I understand from reading the results section, the authors used these assays simply as a qualitative screening. However this should be emphasized in this section, as it is not clear.
- In any case, a description of the methodologies used for the detction of each of these classes of compounds is necessary.
- Finally, a different section should be devoted to the evaluation of the bioactivity of the extracts, in which the description of the methodologies related to Folin and antioxidant activities should be included. I would like to underline that antioxidant assays (ABTS, DPPH, etc ..) cannot be considered analytical assays.
- Why Condensed tannin content (CTC) is actually reported in the paragraph evaluating the antioxidant properties of the extracts?
- Section 2.3. does not report all the chemicals used in the work. Moreover, it should be placed as the first paragraph.
- The meaning of acronyms (eg DPPH, ABTS and HPLC) should be explained in 2.3. section, as they appear for the first time in the text.
- Section 2.6. is inconsistent. Information on the chromatographic gradient, solvents used for separation, and instrumental parameters are not reported. The authors also perform a quantification in HPLC, but no information on the analysis method is reported (did they use standard curves?). Information about the validation of the extraction and analytical process (matrix effect, limit of detection and limit of quantification) are equally not reported.
- From the reported results in Discussion, it seems that the authors limited to a simple UV/Vis analysis. However, the chromatogram shown does not lend itself to analytical quantification without the aid of a mass spectrometer. Please, specify the quantitative analysis method.
in summary, authors should arrange materials and methods as follows:
2.1. Chemicals
2.2. Isolation of Rice Bran (RB)
2.3. Starter Culture for Solid State Fermentation (SSF)
2.4. SSF of RB
2.5. Evaluation of Proximate Composition
- 5.1. Determination of Sugars
- 5.2. Determination of Proteins
2.6. Evaluation of Phytochemical Composition
- 6.1. Determination of Total Polyphenol Content (TPC)
- 6.1. Determination of Saponin
- 6.2. Determination of Steroids
- 6.3. Determination of Polyphenols
- 6.4. Determination of Flavonoids
- 6.5. Condensed tannin Content
- 6.6. Determination of Coumarins
- 6.7. Determination of Alkaloids
- 6.8. Qualitative and quantitative High Performance Liquid Chromatography (HPLC) analysis 110
2.7. Evaluation of Antioxidant Properties of URB and FRB
- 7.1. DPPH assay
- 7.2. ABTS assay
- 7.3. HFRSA assay
- 7.4. Total antioxidant capacity (TAC)
- 7.5. Reducing power assay (RPA)
2.8. Statistical Analysis
RESULTS AND DISCUSSION:
Concerning Results and Discussion, the authors obtained good results. However, I fear these are inconsistent due to lack of specification in the materials and methods section. In particular, a big problem consists in the ambiguity in reporting the data in the tables, where the legends did not help the interpretation of the data.
- table 1 should be renamed based on the observations made previously. In particular, I suggest to specify in the legend “chemical composition” and not “phytochemicals”.
- Moreover, how did the authors establish the absence of some of the compounds? I suppose they rated a limit of detection for each essay.
- Seems very strange that fermented rice has no protein. Are the authors sure of this statement?
- table 2 is very confusing. I would suggest to the authors to place the percentage change in a separate column, and not as a subscript of the numerical value.
- table 3 should be placed in reverse (chemical parameters on the vertical axis, and the samples on the horizontal axis).
- figure 2 shows the UV/Vis chromatogram of the samples. However, the wavelength is not specified in the legend. On what system were the authors' quantifications based? did they use this chromatogram?
- are the values shown in table 5 the IC50 calculated for each assay? specify in the legend of the table.
Author Response
We are extremely thankful to the esteemed reviewers for their valuable comments on the manuscript and to the Editor for giving us an opportunity to improve the manuscript. As desired, we have thoroughly revised the manuscript as per the advises and comments received from the reviewers. The edited portion has been highlighted to make things clearer. Please find below point wise response to the reviewers comments:
Reviewer 1
- No information on the use of Aspergillus oryzae is provided. Aspergillus oryzae is a famous fungus commonly used for the preparation of local foods and beverages in Japan (for example for the preparation of sweet potato, sake, shōchū, soy sauce and miso
Response: Latest work related to fermentation using Aspergillus oryzae has been included in the introduction section.
- Moreover, information regarding other food products based on Rice but produced with other kind of inoculation (for example, fermented red rice) should at least be mentioned, underlining that the methodology operated by the authors is a very common practice aimed at the production of foods with high nutraceutical values.
Response: Needful done. Information is supplemented in the manuscript.
- Section 2 should be renamed: Materials and Methods. This section is really confusing. Materials and methods are poorly described, and some paragraphs are not at all.
Response: As per reviewer suggestion materials and methods section has been improved.
- Subsection 2.5.1. is not clear. What authors state to evaluate in this subsection? No methodology is described in this section.
Response: Detailed methodology has been included in the revised manuscript.
- Moreover, the authors confuse the phytochemical compounds (polyphenols, alkaloids, etc.) deriving from the secondary metabolism of plants, from those deriving from the primary metabolism (sugars, fats, fibres, proteins, minerals). The latter compounds may be included as proximal composition of foods, and not in the phytochemical profile. If all these analyses have been performed, authors should separate them into two different sections (one concerning phytochemical evaluations, and the second one on proximal composition).
Response: As in the present investigation only qualitative tests to confirm the presence of protein and sugars rather than completed proximate composition analysis. Therefore, data related to protein and sugars have been removed from the revised manuscript.
- as far as I understand from reading the results section, the authors used these assays simply as a qualitative screening. However, this should be emphasized in this section, as it is not clear.
Response: Correction incorporated.
- In any case, a description of the methodologies used for the detction of each of these classes of compounds is necessary.
Response: Detailed protocol has been included in the manuscript.
- Finally, a different section should be devoted to the evaluation of the bioactivity of the extracts, in which the description of the methodologies related to Folin and antioxidant activities should be included. I would like to underline that antioxidant assays (ABTS, DPPH, etc ..) cannot be considered analytical assays.
Response: Suggestions have been included in the revised manuscript.
- Why Condensed tannin content (CTC) is actually reported in the paragraph evaluating the antioxidant properties of the extracts?
Response: As suggested by reviewer sequence has been updated.
9.Section 2.3. does not report all the chemicals used in the work. Moreover, it should be placed as the first paragraph.
Response: Major chemicals and standards used in the study has been included.
10.The meaning of acronyms (eg DPPH, ABTS and HPLC) should be explained in 2.3. section, as they appear for the first time in the text.
Response: Suggestions included.
- Section 2.6. is inconsistent. Information on the chromatographic gradient, solvents used for separation, and instrumental parameters are not reported. The authors also perform a quantification in HPLC, but no information on the analysis method is reported (did they use standard curves?). Information about the validation of the extraction and analytical process (matrix effect, limit of detection and limit of quantification) are equally not reported.
Response: Detailed method has been included in the revised manuscript.
Further, all the major changes and sequence of methodology has been updated in the revised manuscript.
- In summary, authors should arrange materials and methods as follows:
Response: Needful done. The sequence of suggested by the reviewer is now updated in the manuscript.
- Concerning Results and Discussion, the authors obtained good results. However, I fear these are inconsistent due to lack of specification in the materials and methods section. In particular, a big problem consists in the ambiguity in reporting the data in the tables, where the legends did not help the interpretation of the data.
Response: We try to more clarify the materials and method section.
- table 1 should be renamed based on the observations made previously. In particular, I suggest to specify in the legend “chemical composition” and not “phytochemicals”.
Response: Title of table 1 has been changed as per reviewer suggestion. Phytochemicals are now replaced with chemical composition.
- Moreover, how did the authors establish the absence of some of the compounds? I suppose they rated a limit of detection for each essay.
Response: The tests performed were based on either color change or formation of precipitates. If the compounds present in sample in amount 0.5µg/ml, in that case there was no color or precipitation.
- Seems very strange that fermented rice has no protein. Are the authors sure of this statement?
Response: Sample tested was extracts rather than pure flour. The protein was absent in extracts during the testing.
- table 2 is very confusing. I would suggest to the authors to place the percentage change in a separate column, and not as a subscript of the numerical value.
Response: As suggested by reviewer, percentage change is now included in separate column.
- table 3 should be placed in reverse (chemical parameters on the vertical axis, and the samples on the horizontal axis).
Response: Suggestion incorporated.
- figure 2 shows the UV/Vis chromatogram of the samples. However, the wavelength is not specified in the legend. On what system were the authors' quantifications based? did they use this chromatogram?
Response: Wavelength used during the HPLC analysis is now included in Figures title.Quantification is based on equation generated from standard compounds curve. Yes chromatogram was used.
- are the values shown in table 5 the IC50 calculated for each assay? specify in the legend of the table.
Response: Rather than IC50 the results are represent on the basis of % Inhibition/%decoloration
Reviewer 2 Report
The article is interesting but needs some minor adjustments.
As a result of the rapid development and changes in civilization, as well as related to them greater stress in humans, there is an increase in interest in the "healthy" mode life and nutrition, as well as interest in synthetic supplements diets. At the same time, it largely deviates from traditional natural, unprocessed foods in favor of processed foods industrially. One such example that was historically large and now has a small share in nutrition, there are fermented products.
Large differences in the course of fermentation of commonly used cereal seeds and their preparations as food products, especially in relation to for acidity, pH as well as antioxidant activity suggest a possibility obtain products for consumption with various properties.
Current knowledge and understanding of antioxidants role in the body tends to pay more attention to their specificity than for the sheer amount of activity
Why did the authors use only one method of testing DPPH antioxidant activity? (ORAC-oxygen radical absorbance capacity, TRAP - total redox antioxidant parameter, CBA - crocin bleaching assay, LPIC - lipid peroxidation inhibition capacity, TOSC - total oxidant scavenging capacity, FCR - Folin-Ciocalteau reagent, ABTS+, CUPRAC - cupric ion reducing antioxidant capacity, FRAP - ferric ion reducing antioxidant parameter).
The negative properties of Aspergillus are also worth describing. The authors did not mention this.
This is not enough for describing a new antioxidant compound
I would recommend to add a little more detail on how the new compounds were identified, having the exact mass, I guess is not sufficient, as most likely more than one suggestion for a molecular formula are provided, just as there are some uncertainty on the measured mass, which also has to be taken into consideration, when deciding on a molecular formula.
I suggest that a diagram presenting the steps of the procedure used in the study be added to the experimental section. It would help understand the details of the analytical protocol better, and allow the written description of the procedure to be shortened
Add citation to materials and methodology:
Line: 109
Grobelna, A., Kalisz, S., & Kieliszek, M. (2019). Effect of Processing Methods and Storage Time on the Content of Bioactive Compounds in Blue Honeysuckle Berry Purees. Agronomy, 9 (12), 860.
Line: 126
Grobelna, A., Kalisz, S., & Kieliszek, M. (2019). The effect of the addition of blue honeysuckle berry juice to apple juice on the selected quality characteristics, anthocyanin stability, and antioxidant properties. Biomolecules, 9 (11), 744.
It is worth for the authors to describe the prospects for the future of their research and possible applications.
Author Response
We are extremely thankful to the esteemed reviewers for their valuable comments on the manuscript and to the Editor for giving us an opportunity to improve the manuscript. As desired, we have thoroughly revised the manuscript as per the advises and comments received from the reviewers. The edited portion has been highlighted to make things more clear. Please find below point wise response to the reviewers comments:
Reviewer 2:
- The article is interesting but needs some minor adjustments.
Response: Thank you for the comments. As per suggestion modifications have been incorporated in the revised manuscript.
- As a result of the rapid development and changes in civilization, as well as related to them greater stress in humans, there is an increase in interest in the "healthy" mode life and nutrition, as well as interest in synthetic supplements diets. At the same time, it largely deviates from traditional natural, unprocessed foods in favor of processed foods industrially. One such example that was historically large and now has a small share in nutrition, there are fermented products.
- Large differences in the course of fermentation of commonly used cereal seeds and their preparations as food products, especially in relation to for acidity, pH as well as antioxidant activity suggest a possibility obtain products for consumption with various properties.
- Current knowledge and understanding of antioxidants role in the body tends to pay more attention to their specificity than for the sheer amount of activity
Response: Thank you for the suggestions. The information mentioned in point 2, 3, and 4 has been updated in the manuscript.
- Why did the authors use only one method of testing DPPH antioxidant activity? (ORAC-oxygen radical absorbance capacity, TRAP - total redox antioxidant parameter, CBA - crocin bleaching assay, LPIC - lipid peroxidation inhibition capacity, TOSC - total oxidant scavenging capacity, FCR - Folin-Ciocalteau reagent, ABTS+, CUPRAC - cupric ion reducing antioxidant capacity, FRAP - ferric ion reducing antioxidant parameter).
Response: We also used ABTS+, Hydroxyl free radical scavenging activity and Total antioxidant capacity method for describing antioxidant activity. These analysis methods were selected on basis of availability of chemicals and facility available in our lab.
- The negative properties of Aspergillusare also worth describing. The authors did not mention this.
Response: Needful done.
- This is not enough for describing a new antioxidant compound
Response: We try our best to use possible methods according availability of facility in our institution.
- I would recommend to add a little more detail on how the new compounds were identified, having the exact mass, I guess is not sufficient, as most likely more than one suggestion for a molecular formula are provided, just as there are some uncertainty on the measured mass, which also has to be taken into consideration, when deciding on a molecular formula.
Response: More information is now added regarding the quantification of bioactive constituents. These analysis were based on standard curve equations.
10 I suggest that a diagram presenting the steps of the procedure used in the study be added to the experimental section. It would help understand the details of the analytical protocol better, and allow the written description of the procedure to be shortened
Response: Detailed Flowchart has been added in the revised manuscript. Citation is added as suggested.
- Add citation to materials and methodology:
Line: 109
Grobelna, A., Kalisz, S., &Kieliszek, M. (2019). Effect of Processing Methods and Storage Time on the Content of Bioactive Compounds in Blue Honeysuckle Berry Purees. Agronomy, 9 (12), 860.
Line: 126
Grobelna, A., Kalisz, S., &Kieliszek, M. (2019). The effect of the addition of blue honeysuckle berry juice to apple juice on the selected quality characteristics, anthocyanin stability, and antioxidant properties. Biomolecules, 9 (11), 744.
Response: Citation is added as suggested.
- It is worth for the authors to describe the prospects for the future of their research and possible applications.
Response: Required information is added.
Round 2
Reviewer 1 Report
Authors followed the suggestions.